# Non-esterified Fatty Acid-Induced Reactive Oxygen Species Mediated Granulosa Cells Apoptosis Is Regulated by Nrf2/p53 Signaling Pathway

**DOI:** 10.3390/antiox9060523

**Published:** 2020-06-14

**Authors:** Yiru Wang, Chengmin Li, Julang Li, Genlin Wang, Lian Li

**Affiliations:** 1College of Animal Science and Technology, Nanjing Agricultural University, Nanjing 210095, China; 2017205015@njau.edu.cn (Y.W.); glwang@njau.edu.cn (G.W.); 2Jiangsu Key Laboratory of Sericutural Biology and Biotechnology, School of Biotechnology, Jiangsu University of Science and Technology, Zhenjiang, Jiangsu 212018, China; chengminli@just.edu.cn; 3Department of Animal Biosciences, University of Guelph, Guelph, ON N1G 2W1, Canada; jli@uoguelph.ca

**Keywords:** non-esterified fatty acid, ROS, *N*-acetyl-l-cysteine, granulosa cells, apoptosis

## Abstract

Negative energy balance (NEB) during the perinatal period can affect dairy cow follicular development and reduce the fecundity. Non-esterified fatty acid (NEFA) concentration is elevated during NEB, and is known to be toxic for multiple cell types. In the ovary, NEB increased NEFA, and may influences follicular growth and development. However, the effect and mechanism of NEFA on granulosa cells (GCs) in vitro remains unknown. In this study, we found that NEFA dose-dependently induced apoptosis in primary cultured granulosa cells. Mechanistically, our data showed that NEFA significantly increased reactive oxygen species (ROS) levels, resulting in the activation of endoplasmic reticulum stress (ERS) and eventually cell apoptosis in GCs. Moreover, NEFA also increased the phosphorylation levels of ERK1/2 and p38MAPK pathways, upregulated the expression of p53 and potentially promoted its translocation to the nuclear, thus transcriptionally activated Bax, a downstream gene of this pathway. NEFA also promoted nuclear factor E2 (Nrf2) expression and its level in the nuclear. To elucidate the mechanism of NEFA action, *N*-acetyl-l-cysteine (NAC), a ROS scavenger was used to verify the role of ROS in NEFA induced apoptosis of GCs. NAC pretreatment reversed the NEFA-induced ERS-related protein and apoptosis-related protein levels. Meanwhile, NAC pretreatment also blocked the phosphorylation of ERK1/2 and p38 induced by NEFA, and the nucleation of Nrf2 and p53, suggesting that ROS plays a crucial role in regulating the NEFA-induced apoptosis of GCs. Together, these findings provide an improved understanding of the mechanisms underlying GCs apoptosis, which could potentially be useful for improving ovarian function.

## 1. Introduction

Due to the integrative outcome of reduced intake and higher demand for maintenance and production, cows in the perinatal period are in the state of negative energy balance (NEB). Non-esterified fatty acid (NEFA) is an important energy metabolite that is used as an indicator of NEB during the perinatal period. The elevation of NEFA is known to result in decreased reproductive potential, especially in high-yielding dairy cows [1]. It was reported that the concentration of NEFA in blood was lower than 0.20 mM in the late lactation and dry milk period, and began to increase in the first two weeks after labor, and the level of NEFA could exceed 0.75 mM at the 10th day of postpartum [2]. In vivo studies have shown that high-yield cows postpartum NEFA level can delay the first ovulation time, increase embryo mortality, reduce the conception rate and pregnancy rate [3]; high-yield cows postpartum NEFA content increase and postpartum first conception rate are negatively correlated, postpartum high-level NEFA cattle conception rate decreased by 50% [4,5]. At the same time, a high level of NEFA can lead to the decrease of insulin-like growth factor (IGF) concentration and affect postpartum ovulation [6]. In addition, a study with 156 lactating cows reported that with the increase of serum NEFA concentration on the third day of postpartum, the pregnancy probability of first artificial insemination decreased [7]. Leroy J et al. [8] showed that high levels of NEFA can reduce the quality of oocytes and embryos. In vitro studies showed that the concentration of NEFA in follicular fluid and serum was consistent with that observed in serum of early lactation cows [9]. However, the high level of NEFA in the follicular fluid will affect the development of cow oocytes in vitro [9]. Serum NEFA level also increased when lipid mobilization increased at the beginning of lactation [10]. Therefore, extensive fat decomposition in perinatal and early lactation not only affects postpartum ovulation but also may affect the function of embryonic oocytes and granulosa cells by changing the lipid composition of follicular fluid [11].

Lipid mobilization is usually accompanied by oxidative stress. The high concentration of NEFA can increase the intracellular production of reactive oxygen species (ROS) and RNS, thus initiating the mechanism of oxidative stress [12]. At the same time, NEFA can damage the structure and function of mitochondria due to its lipid toxicity, which leads to more free radicals such as ROS. On the other hand, oxidative stress can lead to additional fat decomposition, which further increases the concentration of NEFA in perinatal cows [13]. Perinatal cows have to undergo a large range of metabolic and physiological adaptive changes from pregnancy to lactation, which further aggravates the oxidative stress state, and then enter the vicious cycle of fat mobilization and ROS production [14]. In addition, Gessner et al. [15] reported that endoplasmic reticulum stress was prevalent in the liver of perinatal cows. NEFA significantly induced endoplasmic reticulum stress in adipocytes and that the perk pathway can activate IKK β kinase to cause inflammatory injury and insulin resistance in adipocytes [16]. However, whether the high level of NEFA can also induce endoplasmic reticulum stress and apoptosis in ovarian granulosa remains unclear. ROS can activate the signal transduction of NF-κ B, MAPK, and Nrf2, and subsequently induce oxidative stress. Downstream of ROS, p38 and ERK1/2, are the main members of the MAPK signaling pathway. They are activated via phosphorylation and play a role in mediating apoptosis. p38MAPK phosphorylates and activates the Nrf2 transcription factor, Nrf2 is then dissociated from Nrf2/Keap1 binding body, the free Nrf2 transcription factor is then translocated into the nucleus, and regulate the expression of some anti-apoptosis or oxidation genes such as Bcl-2 [17]. It is well known that cellular stress can activate p53 and subsequently affect metabolism, oxidative stress, DNA repair, senescence, and apoptosis [18]. We thus also like to explore the role of p53 and Bax in NEFA induced apoptosis. We hypothesized that NEFA could induce granulosa cell oxidative stress and apoptosis via activating the ROS-MAPK-Nrf2/p53 pathway.

## 2. Materials and Methods

### 2.1. Material and Reagents

Dulbecco’s modified eagle medium (DMEM)/F-12 medium and fetal bovine serum (FBS) were obtained from Gibco Life Technologies (Carlsbad, CA, USA). Oleic acid, linoleic acid, palmitic acid, palmitoleic acid, and stearic acid with 99% purity were from Sigma-Aldrich (St. Louis, MO, USA). N-acetyl cysteine (NAC), RIPA lysate, PMSF, and BCA assay were from Beyotime Biotechnology (Shanghai, China). Anti-ERK1/2, anti-pERK1/2, and anti-α-Tubulin were from Cell Signaling Technology (Danvers, Essex, MA, USA). Anti-Nrf2, anti-HO-1, anti-Bax, -Bcl-2, -cleaved caspase-3, -SOD2, -Histone, -CHOP, -GRP78, -p-PERK, -GAPDH antibodies were purchased from Proteintech (Chicago, CA, USA). The anti-p53 antibody was from Abcam (Cambridge, Cambridgeshire, UK). The secondary antibody and ECL were from Biosharp (Hefei, China). Annexin V- Alexa Fluor 647/PI double-staining kit was purchased from Fumaisi (FMSAV647-100, FcMACS, Nanjing, China). The concentration of stock NEFA solution was 50 mM, containing 21.75 mM oleic acid, 15.95 mM palmitic acid, 7.2 mM stearic acid, 2.65 mM palmitoleic acid, and 2.45 mM linoleic acid.

### 2.2. Primary Culture of Bovine Granulosa Cells

The study was approved by the animal protection and utilization committee (Approval number: SYXK (SU) 2017-0027), Nanjing Agricultural University, Nanjing, China. Briefly, bovine ovaries obtained from slaughterhouses (Nanjing, China) were placed in saline at about 37 °C and returned to the laboratory within 2 h. After washing the ovaries with saline, we used a syringe to extract follicular fluid into saline solution in a sterile environment, centrifuged and suspended cells in complete medium. Subsequently, granulosa cells (GCs) were cultured in DMEM/F-12 medium supplemented with 10% FBS, at 37 °C in a humidified atmosphere of 95% air and 5% CO_2_.

### 2.3. Preparation of Cytoplasm and Nuclear Extracts

The Nuclear and Cytoplasm Protein Extraction Kit (Beyotime, China) was used to extra the nuclear and cytoplasm proteins. Briefly, the cells were collected and 200 μL ice-cold cytoplasm extraction buffer A with 1 mM PMSF was added. After incubation with cytoplasm extraction buffer B for 1 min, cell lysates were centrifuged at 12,000× *g* for 5 min at 4 °C, the supernatant was aspirated as cytoplasmic protein. Next, 50 μL nuclear extraction buffer was added to the sediment. In the next 30 min, vortexing for 30 s every 2 min on the ice took place, then the lysates were centrifuged at 16,000× *g* for 10 min at 4 °C, the supernatant was aspirated as a nuclear protein.

### 2.4. Immunoprecipitation and Immunoblots

According to the instructions for Pierce Co-Immunoprecipitation Kit (Thermo Fisher Scientific, Waltham, MA, USA), the cells cultured in T75 culture bottle were used for immunoprecipitation. For immunoblots, after incubation with NAC and NEFA, the cells were harvested and lysed with RIPA buffer. The protein content was measured by a BCA assay (Beyotime). Equal amounts of protein were resolved using sodium dodecyl sulfate-polyacrylamide gel electrophoresis, then transferred to PVDF membranes. Then, 5% skimmed milk was used to block the membranes, the primary antibodies were incubated at 4 °C overnight. The next day, washing membranes three times in TBST for each 10 min and then incubated with the corresponding secondary antibodies for 1 h at room temperature. Antibody detection was accomplished using enhanced chemiluminescence reagent. Densitometry analysis was detected using ImageJ software (National Institutes of Health, Bethesda, MD, USA).

### 2.5. Flow-Cytometry Analysis

The cells Annexin V-Alexa Fluor 647/PI Apoptosis Assay Kit was used to measure the extent of apoptosis according to the manufacturer’s instructions. Briefly, cells were washed twice with PBS then gently resuspended in binding buffer containing Annexin V/Alexa Fluor 647 (5 uL) and propidium iodide (10 uL) (the cell concentration is 1 × 10^6^/mL). After 15 min in the dark, the apoptosis rates were analyzed by flow cytometry (BD, FACS Calibur, USA). The data analysis was using Flowjo software (Becton, Dickinson and Company, Franklin Lakes, JD, USA).

### 2.6. Reactive Oxygen Species (ROS) Level

The cells were seeded on six-well plates with cover glasses in each well and then treated with NAC for 2 h before treated with NEFA for 24 h. The intracellular levels of ROS were measured by loading the cells with the Dihydroethidium (DHE) (Beyotime Biotechnology, Shanghai, China) [19,20]. The method was based on the DHE, it can enter cells freely through living cell membranes and oxidized by ROS to the product ethidium oxide, which can participate in chromosomal DNA and produce red fluorescence. Briefly, after the treatment, cells were stained with 10 µM DHE in serum-free DMEM-F12 for 30 min at 37 °C in the dark. Then the cells were washed three times with PBS. The fluorescence intensity was measured at 480 nm excitation and 590 nm emission using a fluorescence microscope (Zeiss LSM 700 META (Olympus, Tokyo, Japan)).

### 2.7. Statistical Analysis

All data were expressed as mean ± SEM and the results at least three independent experiments. T-test was used to analyze statistical, using the GraphPad Prism5 software (GraphPad Software Inc., San Diego, CA, USA). P less than 0.05 was considered significant.

## 3. Results

### 3.1. Non-Esterified Fatty Acid (NEFA) Causes Accumulation of ROS, Endoplasmic Reticulum Stress, and Apoptosis in Granulosa Cells (GCs)

To determine the correlation between NEFA and the accumulation of ROS in vitro, granulosa cells were cultured in the absence and presence of 0, 0.6, 1.2, 1.8 mM NEFA for 24 h, and DHE staining was performed. The imaging data showed that ROS signals were upregulated NEFA concentration-dependent manner (Figure 1A). At the same time, the decrease of the SOD2 protein level also suggests the imbalance of redox state (Figure 1B,C). We next studied if NEFA induce endoplasmic reticulum stress. CHOP, p-PERK, and GRP78 are markers for endoplasmic reticulum stress. Western blot analysis revealed that the NEFA upregulated the p-PERK, GRP78, and CHOP protein expression (Figure 1B,D–F) in a dose-dependent fashion.

The incidence of apoptosis was increased in NEFA treated GCs detected via Annexin V-Alexa Fluor 647 staining and flow cytometry (Figure 2A,B). In the meantime, NEFA also prominently increased apoptosis associated Bax/Bcl-2 protein ratio in the GCs (Figure 2C). Taken together, these observations suggest NEFA induces apoptosis to occur in GCs.

### 3.2. Treatment of *N*-acetyl-l-cysteine (NAC) In Vitro Attenuates ROS Levels, Endoplasmic Reticulum Stress (ERS) Levels and Apoptosis in High-Level NEFA Treated GCs

To determine whether supplementation of NAC can decrease ROS, GCs were cultured in the absence and presence of NEFA, and in addition to minor with NAC (5 mM) for 2 h, the levels of the ROS were assessed. As to be expected, immunostaining revealed that NAC substantially reduced the NEFA induced ROS signals (Figure 3A). To further determine if endoplasmic reticulum stress (ERS) could be recovered by NAC in high-level NEFA treated GCs, we next analyzed the expression of p-PERK, GRP78, and CHOP at the protein level. It was found that the levels of p-PERK, GRP78, and CHOP were significantly reduced in the NAC-treated group as compared with the NEFA group (Figure 3B–E).

The occurrences of apoptosis in the NAC-treated GCs were also significantly decreased as compared with those in the NEFA groups (Figure 4A,E). Consistent with this finding, the protein ratio of Bax/Bcl-2 and cleaved caspase-3 also decreased in respond to NAC (Figure 4B–D). These results suggest that ROS induced apoptosis and ERS may be reversed by NAC.

### 3.3. ERK1/2 and p38 Pathways are Involved in NEFA Induced ROS Mediated Apoptosis of GCs

ERK1/2 and p38 are important signaling pathways in mitogen-activated protein kinases, which regulate cell proliferation and apoptosis. To investigate the role of these pathways in NEFA induced apoptosis mediated by ROS, cells were treated with NEFA, and Western blots were performed. As shown in Figure 5A–C, suggesting it may activate the ERK1/2 and p38 signaling pathways. Interestingly, the increased phosphorylation of ERK1/2 and p38 were reversed in the presence of NAC (Figure 5D–F), that ERK1/2 and p38 pathways may be involved in NEFA induced ROS mediated GCs apoptosis.

### 3.4. NAC Eliminates ROS via the Nrf2/HO-1 Pathway in GCs

Nrf2 is an important transcription factor in the regulation of oxidative stress and a central regulator in the maintenance of intracellular redox homeostasis. Heme oxygenase-1 (HO-1) is thought to be a major antioxidant enzyme scavenging cellular ROS [21], and its activation is regulated by Nrf2. Immunoblotting analysis showed that at the concentration of 1.2 mM, NEFA increased the Nrf2 and HO-1 expression at the protein level (Figure 6A–C). To clarify whether the decrease of apoptosis rate by NAC is related to the HO-1 gene, we also examined the protein levels of HO-1 by using immunoblotting. Our results revealed that compare to the NEFA group, NEFA induced HO-1 was significantly reduced by NAC in GCs (Figure 6E,G). Since oxidative stress can activate Nrf2, a classic antioxidant pathway, we next tested whether NEFA induces Nrf2 translocation into the nucleus. Western blot results revealed that nuclear Nrf2 significantly increased in the presence of NEFA, and then this increase was reversed by NAC (Figure 7A). Collectively, these observations suggest that ROS induced by NEFA plays an important role in the activation of the Nrf2/HO-1 pathway.

### 3.5. ROS Induced by NEFA Activated p53/Bax Pathway

The tumor suppressor, p53 was an important transcription factor in response to a diverse number of cellular stress including cell death, oxidative stress, and hypoxia [18]. We next examined the effects of NEFA on p53 expression in GCs. It was found that NEFA increased p53 expression dose dependently (Figure 6A,D). Subsequent study revealed that NEFA induced p53 expression was reversed in the addition of NAC (Figure 6E,F). It is known that upon stressed, p53 translocated to the nucleus, which in turn controls processes such as cell death and metabolism. We next sought to further clarify if NEFA affects p53 translocation. As shown in Figure 7A, an increase of nuclear p53 protein levels was observed in the presence of NEFA. However, pretreating with NAC decreased NEFA induced nuclear p53 (Figure 7A). This suggests that the p53 is translocated into the nucleus in response to NEFA, which may have contributed to NEFA-induced apoptosis and NAC could alleviate this partly.

Recent studies had also shown that when p53 is stimulated, it not only activates cell apoptosis but also interacts with Bcl-2 family members in mitochondria [22,23]. Since our data showed that P53 and Bax accumulated in GCs, we evaluated possible interactions between these two key proteins using immune-precipitation. The results showed that the interaction between P53 and Bax in the NEFA group (Figure 7B,C). In being co-treated with NEFA and NAC group, P53 and Bax interaction was inhibited, suggesting that NEFA, and thus ROS generation may involve in the activation of P53 and Bax interaction.

## 4. Discussion

Recent studies have demonstrated that the follicular fluid total NEFA, ammonia, and urea exceeding the physiological concentration are considered as nutrition and metabolic stressors. Refs. [24,25], which can lead to postpartum reproduction problems [1]. Ovarian granulosa cells play an important role in the reproduction of dairy cows. However, no studies have shown the molecular mechanism of the GCs damage with the high-level NEFA in the perinatal period. High levels of NEFA is known to lead to changes in glucose metabolism by an increase of β-oxidation, resulting in an increase in ROS production [26]. Our findings showed that ROS increases with a dose-dependent manner in NEFA treated cells. Oxidative stress can result in production of reactive oxygen species and nucleophilic substances. If not removed in time, the oxidative stress products will cause oxidative damage, such as cell death, inflammatory response. Our finding that ROS accumulation induced by NEFA mediates apoptosis of GCs, which is consistent with this viewpoint.

The apoptosis response of ovarian granulosa cells to NEFA is concentration-dependent. Some studies showed that oxidative stress and endoplasmic reticulum stress are two processes that cause pathological cell death [27,28]. Our data also showed that there was endoplasmic reticulum stress in NEFA induced apoptosis. More interestingly, our data indicated that providing N-acetylcysteine (NAC), a common antioxidant and scavenger of ROS, attenuated NEFA induced ROS, GRP78, CHOP, and p-PERK protein expression and the apoptosis in GCs. These finding suggest that NAC may prevent NEFA-induced apoptosis, at least partly, via inhibiting ROS generation and ER stress.

Mitogen-activated protein kinases (MAPKs) are important signaling pathways in eukaryotic cell biology, which are involved in the regulation of cell proliferation, differentiation, and apoptosis. ERK1/2 and p38 are members of the MAPK family, their roles in regulating cell survival and other actives were well documented [29,30,31]. It has been reported that p38 MAPK was activated by ROS produced in cells [32]. Our finding that NEFA increase the phosphorylated ERK1/2 and p38 and this increase is block by NAC suggest the NEFA action in GCs may involve ROS activated ERK1/2 and p38 pathways.

The signal transduction process is a very complex network. ROS induced by NEFA can activate ERK1/2 and p38 signal pathways in ovarian granulosa cells, but there is still a lack of clear understanding of the specific process of signal transduction and which upstream and downstream signal molecules are involved. Many studies have shown that the MAPK signaling pathway can activate Nrf2 [33,34,35]. Nrf2 is an important transcription factor that contributes to the expression of antioxidant defense genes. Shen et al. showed that the ERK1/2 pathway up regulate Nrf2 transcription activity [36]. Under damaging stimuli, Nrf2 dissociates from Keap1 and translocates into the nucleus, resulting in the transcription activation of phase II enzyme/antioxidant genes, including HO-1 and GST, etc. [37,38]. We found that the expression of Nrf2 protein increased dose-dependently to NEFA. Our results also showed that under the stimulation of NEFA, nuclear Nrf2 and HO-1 expression were increased, is a cell self-protect response ROS, however, apoptosis still occurred, suggesting that the activation of Nrf2 is not enough to resist the damage caused by oxidative stress. Interestingly, we observed that after pretreatment with NAC, the expression of Nrf2 in the nucleus and its downstream protein HO-1 decreased, that is, NAC did not activate the Nrf2/HO-1 signal pathway successfully. We speculate that the possible reason is that NEFA consumes part of NAC after NAC pretreatment, and the weakened oxidative stress can not compensate for the active Nrf2/HO-1 signal pathway. Previous studies have shown that p38 MAPK activation affects the regulation of downstream transcription factors, such as p53 and Nrf2, to control downstream pro-apoptotic and anti-apoptotic gene expression in response to many extracellular stimuli, including oxidative stress [39,40,41]. p53 is a key player in the process of mitochondria-mediated apoptosis; it enhances the expression of Bax [42]. Our data also showed that high-level NEFA significantly upregulated the expression of p53 and Bax protein and promoted the entry of p53 into the nucleus. After adding NAC to inhibit the accumulation of ROS, the expression of p53 and Bax protein, and the entry of p53 into the nucleus were significantly alleviated compared with the 1.2 mM NEFA group. Our data suggests that the p53 response may be induced by ROS, which is consistent to a previous report [43]. Additionally, our finding on NEFA induced p53 and Bax interaction was suppressed by NAC suggest that NEFA induced p53/Bax signaling pathway activation may be mediated by NEFA induced ROS in granulosa cells.

In summary, our study provides further insights on the molecular mechanisms of NEFA-induced apoptosis of GCs. We found that NEFA induced apoptosis and stimulated an increase in ROS production in a dose-dependent manner in GCs. The accumulation of ROS induced by NEFA results in the up-regulation of ERS related proteins such as p-PERK, GRP78, and CHOP. Importantly, the blockage of ROS production by NAC has significantly reversed NEFA-induced ROS, ERS, and cell apoptosis. NEFA also activated ERK1/2 and p38 pathways, and the effects of NEFA were reversed by pretreatment with NAC. In addition, NAC can also partially reverse NEFA-induced Nrf2 and p53 nucleation and the increased expression of downstream genes HO-1 and Bax to regulate cell apoptosis. Based on our findings, a model of how NEFA induces apoptosis through ROS mediated signaling pathway is proposed (Figure 8). The results of this study may be helpful to further reveal the influence of NEFA on the reproductive function of in the ovary, and to further support the importance of reducing perinatal stress in production.

## Figures and Tables

**Figure 1 antioxidants-09-00523-f001:**
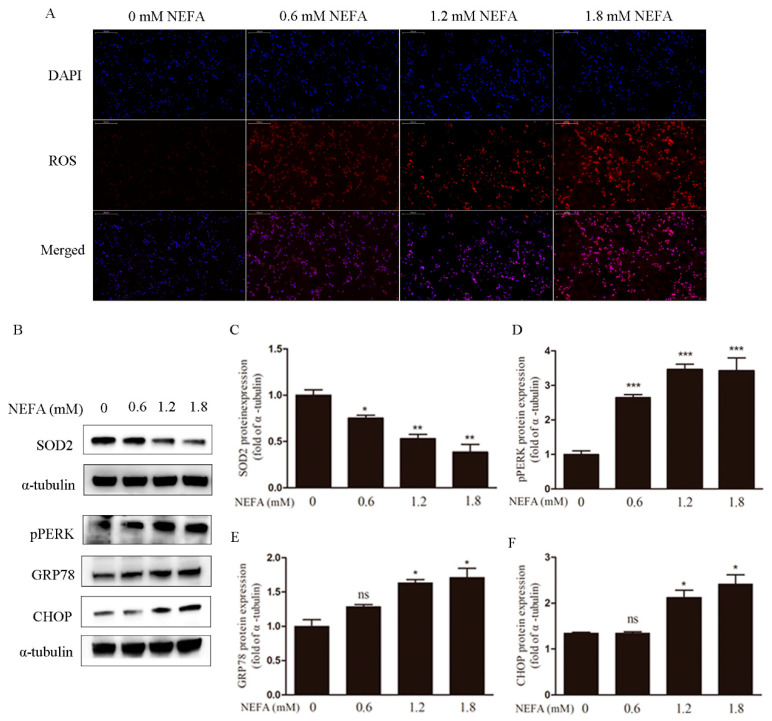
Non-esterified fatty acid (NEFA) causes accumulation of reactive oxygen species (ROS) and endoplasmic reticulum stress in granulosa cells (GCs). (**A**) The accumulation of ROS in different concentrations of NEFA for 24 h was observed by immunofluorescence (scale bar, 200 μm). It indicates that the accumulation of ROS increases with the increase of NEFA concentration. (**B**,**C**) Protein expression of SOD2 was decreased under the different concentrate of NEFA. (**B**,**D**–**F**) Protein expression of p-PERK, GRP78, and CHOP were significantly increased, which means the endoplasmic reticulum stress (ERS) was significantly increased under the high level of NEFA. The results are expressed as the mean ± SEM, * *p* < 0.05, ** *p* < 0.01, and *** *p* < 0.001. ns, not significant.

**Figure 2 antioxidants-09-00523-f002:**
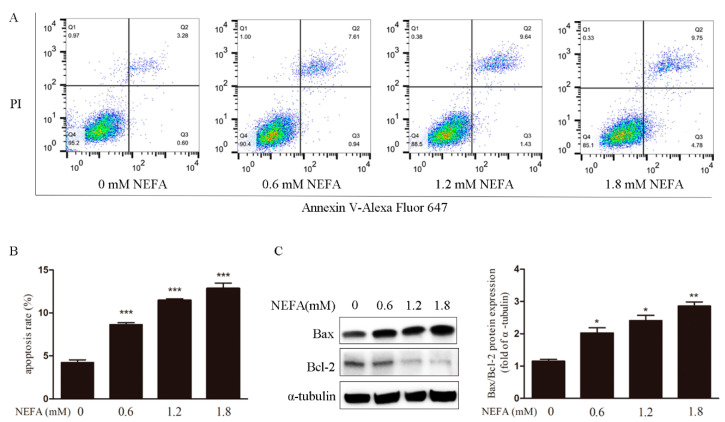
NEFA causes apoptosis in GCs. (**A**,**B**) The percentage of cell apoptosis was determined by Annexin V-Alexa Fluor 647/PI staining and FACS. The results shown are representative of at least three independent experiments. (**C**) The protein level of Bax and Bcl-2 were determined by western blot. The results showed that NEFA-induced apoptosis in GCs. Data are presented as mean ± SEM, * *p* < 0.05, ** *p* < 0.01, and ****p* < 0.001.

**Figure 3 antioxidants-09-00523-f003:**
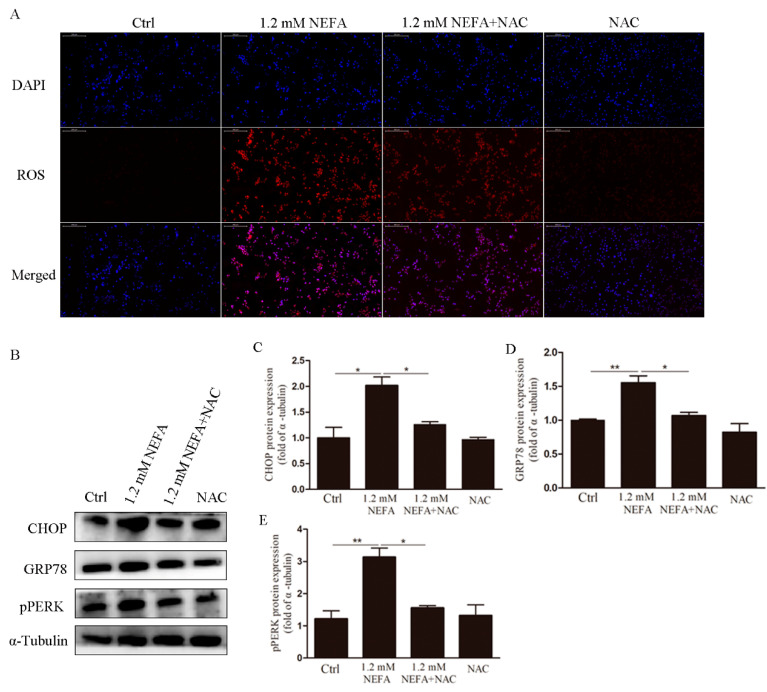
Treatment of *N*-acetyl-l-cysteine (NAC) in vitro attenuates ROS levels, ERS levels in NEFA treated GCs. (**A**) The accumulation of ROS was observed by immunofluorescence (scale bar, 200 μm). The results showed that NAC inhibited the accumulation of ROS induced by NEFA. (**B**–**E**) Western blotting showing protein expression levels of CHOP, GRP78, and p-PERK, which indicate that NAC inhibited ERS. Data are presented as mean ± SEM, * *p* < 0.05, and ** *p* < 0.01.

**Figure 4 antioxidants-09-00523-f004:**
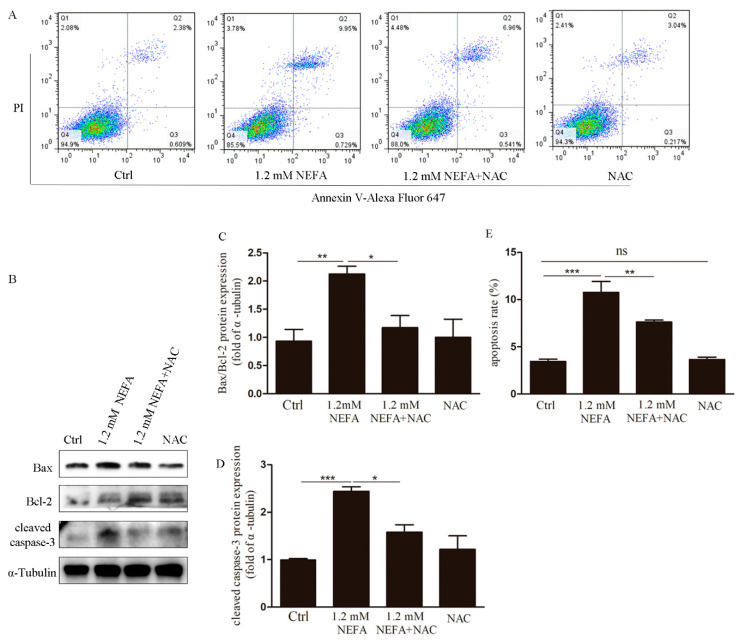
Treatment of NAC in vitro attenuates apoptosis in NEFA treated GCs. (**A**,**E**) Apoptosis events were determined by FACS analysis after Annexin V-Alexa Fluor 647/PI staining in the treated GCs. (**B**–**D**) The results of Bax, Bcl-2, and cleaved caspase-3 protein levels showed that NAC alleviated GCs apoptosis induced by high-level NEFA. Data are presented as mean ± SEM, * *p* < 0.05, ** *p* < 0.01, and *** *p* < 0.001. ns, not significant.

**Figure 5 antioxidants-09-00523-f005:**
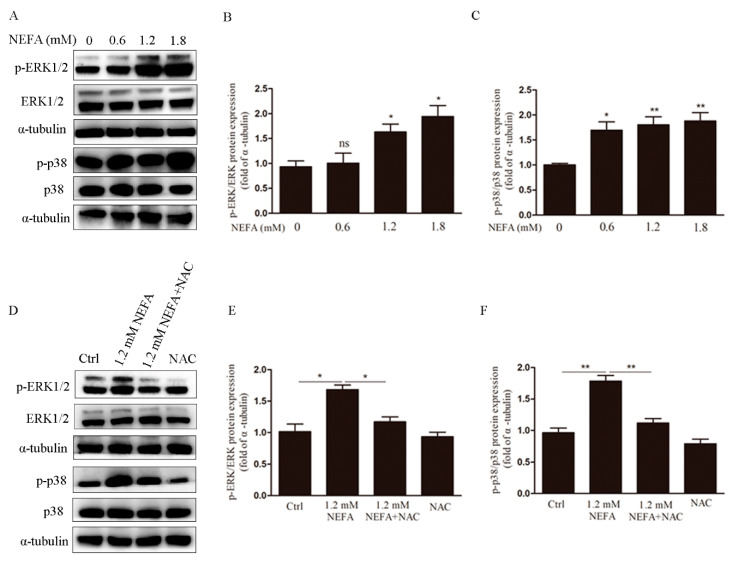
ERK1/2 and p38 pathways are involved in NEFA induced ROS mediated apoptosis of GCs. (**A**–**C**) After treatment with different concentrations of NEFA, p-ERK1/2, ERK1/2, p-p38, and p38 levels were assayed by western blotting. (**D**–**F**) After pre-treatment with NAC, protein expression of p-ERK1/2, ERK1/2, p-p38, and p38 were evaluated by western blotting. These results indicated that the ROS induced by NEFA activated the ERK1/2 and p38 pathways. Data are presented as mean ± SEM, * *p* < 0.05, and ** *p* < 0.01.

**Figure 6 antioxidants-09-00523-f006:**
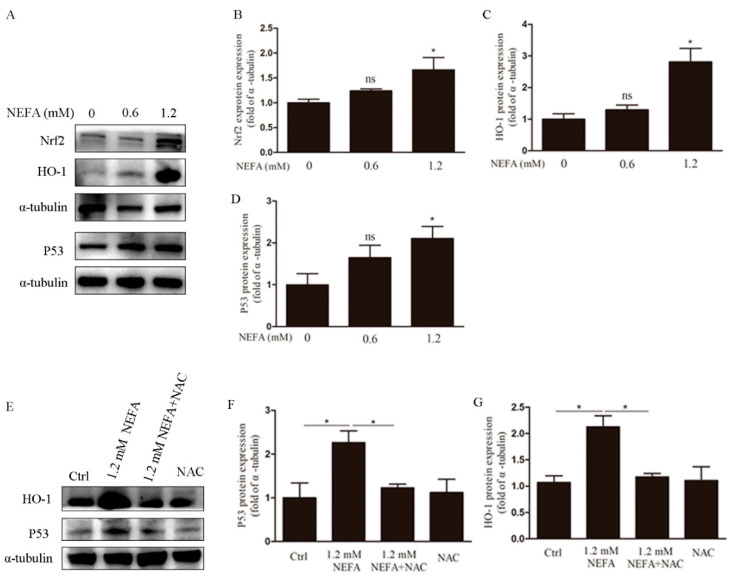
NEFA upregulated Nrf2, HO-1, and p53 protein expression. (**A**–**D**) After treatment with different concentrations of NEFA, protein expression of Nrf2, HO-1, and P53 were evaluated by western blotting. (**E**–**G**) After pre-treatment with NAC, expression of HO-1 and P53 genes were analyzed via western blot. These results indicated that the ROS induced by NEFA upregulated the expression of Nrf2, HO-1, and P53 proteins. Data are presented as mean ± SEM, * *p* < 0.05.

**Figure 7 antioxidants-09-00523-f007:**
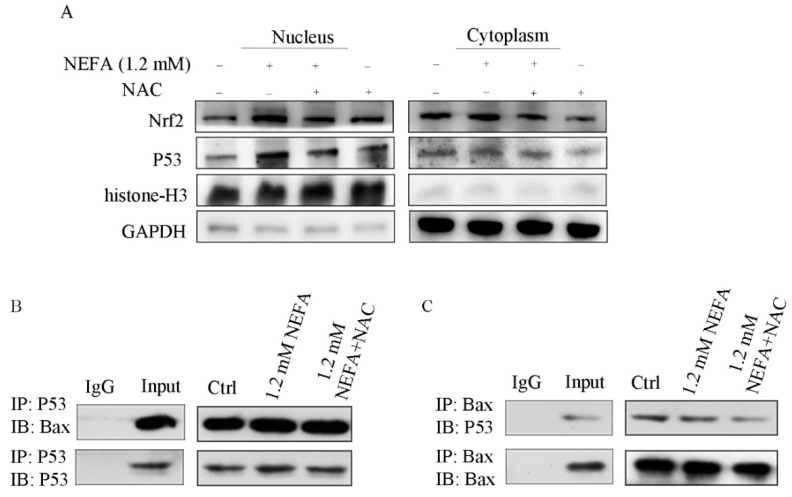
The ROS mediated apoptosis induced by NEFA is regulated by Nrf2/HO-1 and p53/Bax signaling pathways. (**A**) The nuclear levels of Nrf2 and p53 were detected by immunoblotting. (**B**,**C**) The interaction between p53 and Bax is detected by performing co-immunoprecipitation (co-IP) experiments in GCs. The results showed that Nrf2/HO-1 and p53/Bax signaling pathways were involved in NEFA induced apoptosis. “+”, add; “-” without add.

**Figure 8 antioxidants-09-00523-f008:**
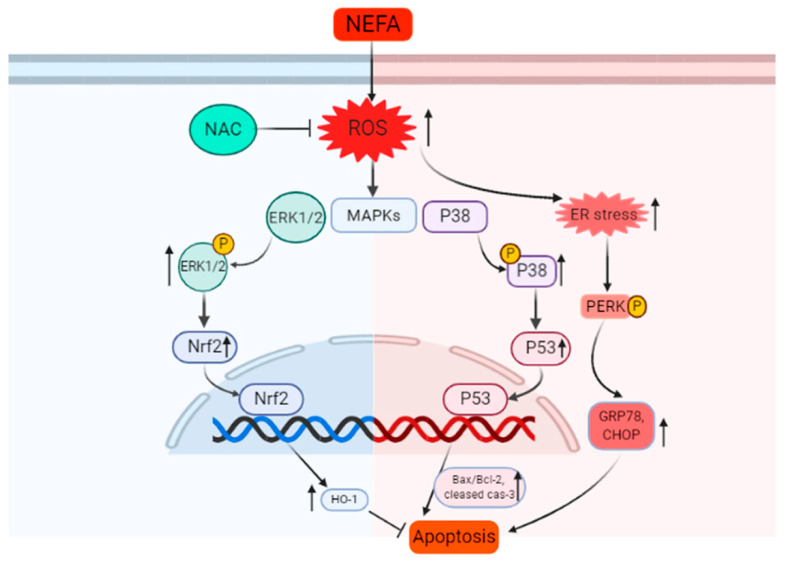
The potential molecular mechanism of NEFA inducing apoptosis in GCs. NEFA induced ROS accumulation, endoplasmic reticulum stress, and apoptosis in granulosa cells. The addition of antioxidant NAC proved that the ROS mediated apoptosis induced by NEFA was regulated by Nrf2/HO-1 and p53/Bax pathway.

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
