# Peer review of "Non-esterified Fatty Acid-Induced Reactive Oxygen Species Mediated Granulosa Cells Apoptosis Is Regulated by Nrf2/p53 Signaling Pathway"

_antioxidants, 2020, doi:10.3390/antiox9060523_

Round 1
Reviewer 1 Report
Review comment to Antioxidant-832602
This investigation studied the functions of non-esterified fatty acid (NEFA) on the apoptosis of granulosa cells and the related mechanism. The result indicated the functions of NEFA on the apoptosis pathway and the coadministration of reducing agents effectively block the apoptosis. The results of this study improved the understanding of the biological functions of NEFA. I recommended accepting this manuscript after addressing the following concern.
- NEFA is a set of compounds from the hydrolysis of triglyceride. I did not see any information on the compound(s) used in this investigation. Which one or sets of NEFA used here should be indicated. If a set of compounds was used, their ratio should be also reported. The structure of those compounds should be presented for a better understanding.
- Regarding the effect of antioxidants, NAC is the only compound used in this experiment and the potential reactions between NAC and NEFA might lead to the observed result. Recommend using other antioxidants for these experiments.
- In figure 2, the ratio of BAX/Bcl-2 was used as an indicator. However, both are proteins related to apoptosis, why not use alpha-tubulin as reference?
- In the discussion, the authors claim that NAC could “reverse” the alteration of NEFA on ERK and P38 (line 375-277). However, the experiment was carried with the co-administration of NEFA and NAC. NAC “inhibits” or “block” the damage caused by NEFA should be a better statement. Such a statement with “reverse” might confusing readers
- Almost all the figures need to improve. It is very difficult to read with the current format. Enlargement is necessary.
Author Response
Dear reviewer,
Thanks very much for your value advice!
Please see the attachment to view the response to the comments. Thanks!
Yiru Wang

Reviewer 2 Report
Please correct some typos thoughout the manuscript.
The authorization number of animal experimental procedure has to be included in the material and method section.
The quality of figures should be improved. Basically, it is very difficult to read. Please, enlarge them.
The author contributions section should be edited according to journal formatting rules.
Author Response
Dear reviewer,
Thanks for your comments. Please see the attachment of the response.
Have a good day!
Yiru Wang

Reviewer 3 Report
Manuscript number Antioxidants-832602
entitled: Non-esterified fatty acid-induced reactive oxygen species mediated granulosa cells apoptosis is regulated by Nrf2/p53 signaling pathway
This is valuable and a well conducted scientific study, done in a thorough manner and expressed concisely. Therefore, the manuscript is suitable for Antioxidants after considering the below comments:
- Line 102, 104 and 113 please add space to 4space°C
- Line 97 and 131 please add space to 37space℃
Author Response
Dear reviewer,
Many thanks for your careful review and kindly suggestions!
Response:
- Thanks. I have corrected as required, please see line 105, 107, 116.
- Thanks. I have added space in line 99 and 134, please check it.
Thanks again for the time and effort you have spent in our manuniscript.
Yiru Wang